# Lymphocyte T Subsets and Outcome of Immune Checkpoint Inhibitors in Melanoma Patients: An Oncologist’s Perspective on Current Knowledge

**DOI:** 10.3390/ijms25179506

**Published:** 2024-08-31

**Authors:** Clara Martínez-Vila, Europa Azucena González-Navarro, Cristina Teixido, Roberto Martin, Francisco Aya, Manel Juan, Ana Arance

**Affiliations:** 1Department of Medical Oncology, Althaia Xarxa Assistencial Universitària de Manresa, Dr. Joan Soler, 1–3, 08243 Manresa, Spain; cmartinezv@althaia.cat; 2Programa de Doctorat en Medicina i Recerca Translacional, Facultat de Medicina, Universitat de Barcelona, 08036 Barcelona, Spain; 3Institut de Recerca i Innovació en Ciències de la Vida i de la Salut a la Catalunya Central (IRIS-CC), Roda 70, 08500 Vic, Spain; 4Department of Immunology, Hospital Clínic of Barcelona, University of Barcelona, Villarroel 170, 08036 Barcelona, Spain; 5August Pi i Sunyer Biomedical Research Institute (IDIBAPS), Rosselló 149, 08036 Barcelona, Spain; 6Department of Pathology, Hospital Clínic of Barcelona, University of Barcelona, Villarroel 170, 08036 Barcelona, Spain; 7Department of Medical Oncology, Hospital Clínic of Barcelona, University of Barcelona, Villarroel 170, 08036 Barcelona, Spain; 8Grupo Español de Terapias Inmunobiológicas en Cáncer (GETICA), Velázquez 7, 28001 Madrid, Spain

**Keywords:** melanoma, anti-PD-1, anti-CTLA-4, lymphocyte T subsets, biomarkers

## Abstract

Melanoma is the most aggressive and deadly form of skin cancer, and its incidence has been steadily increasing over the past few decades, particularly in the Caucasian population. Immune checkpoint inhibitors (ICI), anti-PD-1 monotherapy or in combination with anti-CTLA-4, and more recently, anti-PD-1 plus anti-LAG-3 have changed the clinical evolution of this disease. However, a significant percentage of patients do not benefit from these therapies. Therefore, to improve patient selection, it is imperative to look for novel biomarkers. Immune subsets, particularly the quantification of lymphocyte T populations, could contribute to the identification of ICI responders. The main purpose of this review is to thoroughly examine significant published data on the potential role of lymphocyte T subset distribution in peripheral blood (PB) or intratumorally as prognostic and predictive of response biomarkers in advanced melanoma patients treated with ICI regardless of BRAFV600 mutational status.

## 1. Introduction

Since the approval of ICI, which targets Cytotoxic T-Lymphocyte Antigen 4 (CTLA-4) and Programmed cell Death 1 (PD-1), the course of treatment and prognosis for metastatic melanoma (MM) have undergone significant progress. Both belong to the CD28 family and function as inhibitory receptors that impede the activation and proliferation of T lymphocytes. However, they intervene at different phases of the T cell activation process [1]. During the early phase of T cell activation, naive T cells in the lymph nodes express CTLA-4, which binds to the costimulatory protein B7 produced by antigen-presenting cells (APCs). This prevents the costimulatory signal CD28 from binding, thereby blocking the activation of naive T cells. On the other hand, the PD-1 receptor, expressed by an already activated circulating T cell or present in inflamed tissue, interacts with its ligand, PD-L1, expressed by other cell types such as tumor cells or T cells. Once this interaction takes place, a deactivation of T cells occurs, which prevents an effective immune response at the peripheral level [1]. PD-1 is moderately expressed on naive T, B, and NK cells, upregulated in response to T and B cell receptor activation, and induced on activated monocytes and myeloid cells.

Phase III clinical trials have shown that ipilimumab, a human IgG1 monoclonal antibody that blocks CTLA-4, significantly prolongs progression-free survival (PFS) and overall survival (OS) in patients with advanced melanoma when compared to gp100 vaccination [2] or dacarbazine (DTIC) [3], with an overall response rate (ORR) of 11%, a median PFS of 2.9 months (mo), and a median OS of 10.1 mo. Subsequently, pembrolizumab and nivolumab, both anti-PD-1, were approved as first-line anti-PD-1 monoclonal antibodies for the treatment of advanced and/or MM. The approval was based on the results of pivotal trials in which anti-PD-1 monotherapy achieved an objective response rate (ORR) of approximately 35 to 42% and a 7-year OS of 37.8%, demonstrating superiority over chemotherapy (ChT) and anti-CTLA-4 as first-line treatment in patients with advanced melanoma [4,5].

Later on, data from the pivotal studies of the combination of anti-PD-1 and anti-CTLA-4 were reported, which have been shown to act synergistically, achieving ORR superior to anti-PD-1 monotherapy of 50–70%, with also greater OS at 5 years than monotherapy, around 59–68% [6,7] varying between studies, even in patients with central nervous system (CNS) involvement [6,7]. Mechanisms underlying the greater efficacy of combination therapy may include enhanced priming and activation of helper and cytotoxic T cells and regulatory T cell clearance mediated by CTLA-4 blockade, together with restored cytotoxic T cell killing activity induced by both PD-1 and CTLA-4 inhibition [7].

Another emergent combination of ICI that has shown superiority to anti-PD-1 monotherapy is anti-PD-1 (nivolumab) plus anti-LAG-3 (relatlimab). Updated results with a 19 mo median follow-up showed a median PFS of 10.2 months in comparison to 4.6 mo with nivolumab monotherapy; median OS was not reached (NR) versus 34.1 months with nivolumab alone; ORRs were 43.1% with the combination in comparison to 32.6% with nivolumab monotherapy [8].

Although both combination regimens, nivolumab plus ipilimumab and nivolumab plus relatlimab, have not directly been compared as first-line treatment in advanced melanoma patients, grade ≥ 3 toxicity due to iRAE has been reported to be much lower with the second combination (21.1% vs. 40–60%) [8].

On the other hand, selective BRAF inhibitors (BRAFi) have demonstrated remarkable clinical activity in melanoma patients carrying BRAFV600 mutations [9], which improved with the combination with MEK inhibitors (MEKi). The combination of BRAFi and MEKi using three distinct combinations has achieved an ORR of 64–69% and a 5-year OS rate of around 34% [9,10,11,12] with distinct toxicity profiles. However, BRAF/MEKi are not the topic of this review; therefore, this treatment option will not be further discussed.

The effectiveness of ICI in treating melanoma is due to its highly immunogenic nature [13]. The major reason for the considerable immunogenicity is generally attributed to a substantial number of tumor mutations, mostly resulting from UV-induced alterations in the genetic material. This leads to the production of neoantigens, which are identified as foreign by the immune cells of the host, causing their activation [14]. Even so, suboptimal activation of tumor antigen-specific T lymphocytes, resulting from low or absent expression of these antigens or molecules involved in their presentation [15], may inhibit T cell homing to the tumor [16]. Even in the case of optimal T cell activation, problems may also arise from the lack of appropriate chemoattractants in the tumor area or from their receptors on T lymphocytes.

According to the literature, various immune populations can either enhance or hinder an effective immune response [13,14,15,16]. These immune populations, in particular lymphocyte T cells that originate from the lymphoid lineage, can be quantified in peripheral blood (PB) isolating PBMCs or intratumorally, as tumoral infiltrated lymphocytes (TILs), through flow cytometry or more advanced techniques such as single-cell RNA sequencing (scRNAseq), mass cytometry, or, in the case of TILs, indirectly determined with immunohistochemistry (IHC). For instance, studies on melanoma patients have shown that a high density of activated T cells surrounding the tumor or an elevated count of B lymphocytes in the same location are associated with improved outcomes [17,18]. Furthermore, because of the immunogenic nature of melanoma, analyzing the immune infiltrate components might serve as biomarkers to predict the effectiveness of treatment in patients receiving ICI. This is in addition to its prognostic usefulness [13,14,15,16].

Furthermore, PD-L1, the ligand for PD-1 expressed by tumor and inflammatory cells, is also considered a biomarker of immune status. It mediates T cell deactivation by inhibiting T cell proliferation and cytotoxic potential, as well as cytokine production, resulting in less functional and inactivated T cells, known as exhausted T cells [19,20]. Higher PD-L1 expression by IHC has been extensively associated with better anti-PD-1 outcomes [19,20]. However, a lack of PD-L1 expression does not necessarily preclude a response, particularly in melanoma [19,20]. Therefore, IHC detection of PD-L1 as a predictive biomarker for ICI outcome in melanoma has limited efficacy.

The present review focuses on significant published data that explore the usefulness of intratumoral or circulating lymphocyte T subset determination in advanced melanoma patients treated with ICI, regardless of BRAFV600 mutational status, as prognostic or predictive of response biomarkers.

Particularly, we will center on anti-CTLA-4 and anti-PD-1 monotherapy or anti-PD-1 in combination with anti-CTLA-4 within ICI treatments since it is the most explored and has the most scientific evidence about lymphocyte T population analysis. We will use the anti-PD-1-based therapy concept on several occasions, which encompasses anti-PD-1 monotherapy, anti-PD-1 in combination with anti-CTLA-4, and anti-PD-1 in combination with anti-LAG-3.

## 2. Results

### 2.1. Lymphocyte T Populations and Outcomes of Melanoma Patients Treated with Anti-PD-1 or Anti-CTLA-4 Monotherapy and Anti-PD-1 Plus Anti-CTLA-4 Combination

Although the functional differences between T cell subpopulations are also reflected in gene expression, chromatin, metabolism, and ultimately, at the functional level, in this review, we will focus on T cell surface-expressed antigens and intracellular proteins. Historically, multiparameter flow cytometry has been useful in identifying new subsets of T cells with a defined set of specialized markers and functions [21]. Nowadays, the development of innovative high-content technologies, such as scRNA-seq or mass cytometry, is redefining the phenotypic classification and functional landscape of T cells [21]. Due to the diversity of the human immune system, there is a significant initiative called the Human Immunology Project. Its goal is to establish standardized protocols for all flow cytometry and more modern methods, as already cited [22]. T cells consist of significant subgroups, including CD4^+^ and CD8^+^. However, improved scientific knowledge has proven that certain cells within these subsets possess plasticity and can differentiate into various subpopulations.

In melanoma, some studies have reported that a prominent lymphocytic infiltrate is an independent parameter associated with a good prognosis [14], particularly when lymphocytes infiltrate the tumor in the vertical growth phase [23]. In particular, T lymphocyte infiltration, especially CD8^+^ and CD45RO^+^ subsets, has been shown to be a positive prognostic factor in several cancer types [24]. However, activated T cells in the tumor microenvironment are often functionally defective, with reduced proliferative capacity, cytokine production, or cytotoxicity [16]. Therefore, when assessing their role as biomarkers, it is important to characterize their functional state in addition to their quantitative determination in PB or TILs, for example, by measuring ki67 expression.

#### 2.1.1. Memory Compartment from the Lymphocyte T Cell Population

Naive (Tn), Central memory (Tcm), Effector memory cells (Tem), Terminally differentiated effector T cells (Temra), and Effector T cells (Teff).

Hematopoietic stem cells give rise to different types of blood cells, essentially two main lineages: myeloid and lymphoid. T cells, B cells, and NK cells arise from the lymphoid lineage (Figure 1). After maturation in the thymus, where they undergo somatic gene rearrangement resulting in the expression of a unique T cell receptor (TCR), T cells are fated to a particular CD8^+^ or CD4^+^ lineage. This occurs through a process called positive selection, which involves the recognition of major histocompatibility complex class 1 (MHC1) or class 2 (MHC2) antigens corresponding to CD8^+^ or CD4^+^ T cell subsets, respectively (Figure 1) [25]. In the case of a self-antigen presentation, this process enables and ensures the elimination of autoreactive T cells [25]. Ultimately, remaining positively selected T cells enter the circulation, patrolling the blood and secondary lymphoid organs (SLOs), such as the spleen and lymph nodes, displaying a CD45RA^+^CCR7^+^ naïve T cell phenotype (Tn) [21,25]. Afterward, during their journey, if they encounter APCs carrying their cognate antigen, the Tn cells bind to it via the TCR in the presence of a sufficient co-stimulatory signal. Once this co-stimulation process occurs, Tn cells are activated and differentiate into effector cells (Teff), which migrate to sites of inflammation and attack infected cells [21]. Once the pathogen or other threat, such as tumor cells, has been eliminated, 95% of the effector cells undergo apoptosis. Surviving T cells form a memory fraction that can recognize an antigen upon re-exposure, presenting a phenotype with a loss of CD45RA and an increase in CD45RO (Figure 2) [21]. We can identify CCR7^+^ central memory (Tcm), CCR7^−^ effector memory (Tem), and CCR7^−^ effector memory T cells that re-express CD45RA (Temra) [21]. In peripheral organs, Temra and Tem detect antigens, trigger a fast cytokine response, and release cytotoxic chemicals to kill antigen-expressing cells [21]. On the other hand, Tn and Tcm cells migrate through the SLOs, where they can be activated by mature dendritic cells, undergo expansion, and develop into Teff and Tem cells. These cells then circulate in the peripheral tissues through the bloodstream [21].

In the localized setting, a study that analyzed PB T cell distribution in a cohort of 35 patients with stage resectable III to IV melanoma treated with ipilimumab in a neoadjuvant setting showed an increased frequency of memory T cells expressing activating cytokines but no increase in Tn cells after 6 weeks of treatment (*p* < 0.05) [26] (Table 1).

In recurrent and advanced settings, similar observations have been made. For instance, in the analysis of PB using flow cytometry from patients included in ipilimumab pivotal trials (CA184-004 and CA184-007), there was an increase in Tem populations after four weeks of treatment, while Tn CD8^+^ and CD4^+^ cells declined [27].

Also, in a study involving 77 patients with advanced melanoma treated with ipilimumab [28], an early increase in Tcm and Tem subsets was observed in PB samples from patients with disease control (DC). In addition, patients with DC had a slightly delayed but sustained increase in CD8^+^ Tem cells (*p* < 0.05). The same study [28] assessed the dynamics between memory T cell subsets based on the expression of the Ki67 protein. Ipilimumab treatment resulted in a significant increase in CD4^+^ and CD8^+^ T cell proliferation (higher Ki67 expression) (*p* < 0.05), involving the Tcm and Tem subsets while remaining barely detectable in Tn.

Further evidence supporting that Tem and Tcm correlate with ICI response was observed in another study including 137 patients with advanced melanoma treated with ipilimumab. In this study, those patients with 13% or higher percentages of Tcm CD8^+^ in PB were longer responders to treatment [29].

Furthermore, in another study analyzing flow cytometry data with supervised clustering from PBMCs in 90 advanced melanoma patients treated with anti-PD-1 and anti-CTLA-4 combinations, an increase in circulating CD4^+^ Tem cells correlated positively with longer PFS [30].

In addition, high-dimensional single-cell mass cytometry was used for in-depth characterization of immune cell subsets in PB and matched tumor biopsies of 20 patients with MM before and after 12 weeks of anti-PD-1 treatment [31]. Patients classified as responders had higher numbers of CD4^+^ and CD8^+^ TILs, a significantly lower frequency of circulating CD4^+^ Tem cells, and a lower frequency of CD8^+^ Tn cells at baseline and after treatment initiation (*p* < 0.05). In addition, the CD8^+^ T cell subpopulation of responders had a higher frequency of Tcm cells before and after the start of treatment than the CD8^+^ T cell subpopulation of non-responders [31].

Moreover, Ribas and colleagues [32] reported an increase in memory T cells (CD8^+^CD45RO^+^) in tumoral samples from melanoma patients who presented a good outcome with pembrolizumab treatment. Also, flow cytometry from PBMCs isolated from 31 melanoma patients treated with anti-PD-1 or anti-PD-1 plus an anti-CTLA-4 combination revealed that responders showed an increase in the CD8^+^ Tem subset after three weeks of treatment [33].

Importantly, recent studies suggest that the level of CX3CR1, a chemokine receptor and marker of T cell differentiation within memory subsets, could behave as a predictor of response to anti-PD-1 therapy. According to a study published in Nature [34], administering anti-PD-1 to tumor-bearing mice increased the frequency and clonality of the T cell receptor belonging to the CX3CR1^+^CD8^+^ subset. The same study observed a higher frequency of the CX3CR1^+^ subset on circulating CD8^+^ T cells shortly after treatment initiation in patients with non-small-cell lung cancer (NSCLC) treated with anti-PD-1. ORR, PFS, and OS correlated positively with this frequency. Primarily, CX3CR1 is irreversibly expressed once T cells are fully differentiated; thus, in this study, its upregulation was maintained during treatment, but Ki67 expression on CD8^+^ T cells peaked at day 14 and returned to baseline by day 21 [34].

Finally, several studies have increased interest in the Tcm/Teff ratio as a predictor of ICI response. In this regard, in a study of patients with irresectable and/or metastatic melanoma (MM) (*n* = 43) and NSCLC (*n* = 40) treated with nivolumab [35], tumors with increased inflammatory gene transcripts had increased ratios of CD4^+^ and CD8^+^ Tcm cells to Teff in the blood. In addition, the Tcm/Teff ratio correlated positively with PFS. The study [35] hypothesizes that a higher Tcm/Teff ratio does not reflect the immune response against the tumor but rather the individual’s ability to mount this response, which can be potentiated by ICI, particularly anti-PD-1.

Interpreting all these results, it seems that regardless of the type of ICI, anti-PD-1 or anti-CTLA-4 monotherapy or anti-PD-1 plus anti-CTLA-4, the populations of Tcm and Tem, especially within the CD8 compartment, are expanded after treatment, and their higher frequency in PB and TILs within the total T cell distribution correlates with the magnitude of the treatment benefit. In contrast, a decrease in Tn, both in the CD4^+^ and CD8^+^ compartments, is observed after ICI treatment, and this could confer a better prognostic in melanoma patients, as it was observed in stage III patients. Also, the CX3CR1 chemokine could be crucial in mediating memory subset maturation upon anti-PD-1 treatment.

##### Resident Memory T Cells (Trm)

After migrating to different lymphoid and non-lymphoid tissues, some of the memory T cells generated from persistent activation against specific antigens eventually acquire tissue-resident properties characterized by up-regulation of CD69 and CD103, which are hallmarks of tissue-resident memory T cells (Trm). The Trm subpopulation persists long-term in tissues without expanding into PB and acts as a sentinel in the event of antigen re-exposure [36] (Figure 2).

These cells have many intrinsic properties that imply they could regulate the growth of tumors: (i) they react far faster than circulating memory cells to re-exposure to cognate antigens; (ii) they express large amounts of cytotoxic molecules; and (iii) they are highly expressed in the tumor microenvironment. Trm are found in many human cancers, including melanoma, and they share all the features of memory CD8^+^ T cells (gated as CD45RO^+^CD62L^−^CD28^−^CD27^−^CCR7^−^) as well as an antitumor role in general [37]. They are the lymphocyte population with more immune checkpoint molecule expression [38], and preliminary data have shown that they may expand early during anti-PD-1 treatment.

A study examined the functional profiles of T cells within individual lymph metastases in 44 patients with stage III melanoma using single-cell mass cytometry undergoing anti-PD-1 adjuvant treatment. The outcomes were contrasted with circulating memory-phenotypic T cells, which were mainly CD8^+^ T cells [39]. The percentage of intratumoral T cells that expressed checkpoint inhibitory proteins, PD-1 and TIM-3, was greater than that of circulating T cells. Nearly 60% of CD8^+^ T cells were found to be CD45RO^+^CD69^+^CCR7^−^, which is consistent with the Trm cell phenotype. Moreover, in the same study, a greater quantity of tumor-resident CD69^+^CD103^+^CD8^+^ T cells was linked to better melanoma-specific survival in stage III patients not receiving adjuvant anti-PD-1 [40].

Wei et al. showed that T cell clones expanding in PB during anti-PD-1 treatment expressed high levels of CD69, PD-1, LAG-3, and CD45RO, corresponding to the phenotype of the CD8^+^ Trm population [41].

Interestingly, it is also worth mentioning that in a study including 103 resectable stage III melanoma patients, the presence of CD39^+^ Trm in the surgical margin was associated with better relapse-free survival (RFS) [42].

Apparently, Trm are expanded after anti-PD-1, and their higher frequency correlates with a better prognosis in melanoma patients. However, they are not found in peripheral blood; thus, invasive methods are needed for their determination intratumorally.

#### 2.1.2. CD4^+^ T Cells

For CD4^+^ T cells to identify an antigen, it must be delivered via MHCII, which is only expressed by specialized APCs such as dendritic cells (DCs), B cells, and macrophages [16]. When these APCs identify pathogens or other agents, such as tumor cells, that they recognize as danger signals, they become activated and migrate to the SLOs. There, they are dedicated to displaying proteins and peptides derived from pathogens on their surface and presenting them to different types of cells, among them CD4^+^ T cells, which need to recognize MHCII to cause their activation. At the time of activation of the CD4^+^ T cells, depending on which cytokines are present in the environment, differentiation towards a specific subtype takes place, allowing a more efficient response according to the type of pathogen. In the presence of the cytokines interleukin (IL)-12 and interferon-gamma (IFN-γ), they induce the expression of the transcription factor T-bet in T cells, resulting in T helper type 1 cells (Th1) producing IFN-γ, which at the same time contributes to the death of the pathogen. In contrast, if IL-4 and IL-2 are present, IL-4 will induce Gata3 expression, resulting in helper type 2 (Th2) cells that secrete IL-4, IL-5, and IL-13, critical mediators of extracellular parasite clearance [16].

However, in recent years, the spectrum of CD4^+^ T cell subsets has been broadened, with the identification of T helper cell types 17 and 22 (Th17, Th22) and regulatory T cells (Treg). Th17 cells are generated by the expression of RORγt and secrete IL-17. IL-17, in turn, stimulates antimicrobial defense at the epithelial level and results in the recruitment and activation of neutrophils [16]. Th22 cells exert their effects on epithelial cells by releasing IL-22, which facilitates wound healing and provides tissue protection against injury [16]. A comprehensive examination of CD4^+^ T cell subsets has demonstrated distinct migratory capacities, as indicated by the presence of a specific group of chemokine receptors. The chemokine receptors CCR4, CCR6, CCR10, and CXCR3 have been identified as distinctive combinations for each CD4^+^ T cell subtype. These combinations are as follows: Th1 (CCR6^−^CCR4^−^CXCR3^+^), Th2 (CCR6^−^CCR4^+^CXCR3^−^), Th17 (CCR6^+^CCR4^+^CXCR3^−^CCR10^−^), and Th22 (CCR6^+^CCR4^+^CXCR3^−^CCR10^+^) [16].

Importantly, the CXCR3 receptor interacts with CXCL9, CXCL10, and CXCL11 chemokines, which are released in the presence of IFN-γ and attract CXCR3-positive cells such as Th1 to sites of inflammation [16] (Figure 3).

Conversely, Forkhead Box P3 (FoxP3)^+^ Treg cells do not aid in the defense against pathogens but instead act to prevent autoimmune disorders by inhibiting undesired immune responses [16].

##### T Helper (Th) Subset

Although the specific roles of different subsets of CD4^+^ T cells in various types of tumors are not well understood and require more investigation, it is widely acknowledged that Th1 cells enhance antitumor immune responses by producing significant quantities of IFN-γ. This not only promotes the function of cytotoxic CD8^+^ (Tc) T lymphocytes but also attracts classically activated NK cells and M1 macrophages [43]. Nevertheless, within the framework of MM, there is a lack of definitive research that establishes the prognostic significance of evaluating CD4^+^ TIL by histopathology [15]. A positive association between higher densities of CD3^+^ and CD8^+^ TIL and OS has been observed, but not for CD4^+^ TIL [44]. In vivo murine models, CD4^+^ Th1 and Th2 cells have been shown to efficiently eliminate B16 melanomas [45].

Higher expression of Th1-associated genes, such as tumor necrosis factor-alpha (TNF-α) and IL-2, was detected in 20 primary melanoma tumors that spontaneously regressed, indicating activation of the host antitumor immune response, compared with non-regressing tumors [46].

In a phase II trial [47], which included 75 patients with high-risk resected melanoma, HLA-A*0201-positive patients received a peptide vaccine in combination with ipilimumab. In this population, increased peripheral blood Th17 frequencies 6 months after treatment initiation were correlated with RFS (*p* < 0.05).

Additionally, a study evaluated whether PD-1 blockade influenced T-helper polarization. To do this, whole blood and PBMCs from 9 patients with MM, 15 patients with prostate cancer, and healthy controls were confronted with a superantigen (staphylococcus enterotoxin B) and re-exposure to the tetanus toxoid antigen to evaluate the reactivity of T cells [48]. Adding anti-PD-1 to the medium was observed to change the polarization of CD4^+^ T cells reacting to these antigens towards Th1 and Th17 phenotypes based on increased IFN-γ, IL-2, TNF-α, IL-6, and IL-17 production and the decrease in cytokines in the environment typically produced by Th2, IL-5, and IL-13.

However, two studies evaluating differences between anti-PD-1 plus anti-CTLA-4 combinations in comparison to anti-PD-1 and anti-CTLA-4 monotherapies [41,49], which will be further discussed in Section 2.3, reported that Th1 population expansion occurred after anti-CTLA-4 alone and even more with the combination but was not observed with anti-PD-1 monotherapy.

Taking all this information into account, Th1 and Th17 have an antitumor role, and both seem to increase after anti-CTLA-4 treatment. There are controversial results about Th1 expansion after anti-PD-1 monotherapy, but importantly, it seems that anti-PD-1 and anti-CTLA-4 combinations potentiate Th1 population expansion. Additionally, an increase in both Th populations seems to correlate to a better prognosis in melanoma patients. Although no clear association with responsiveness to anti-PD-1 has been observed, there are data about Th17 expansion and better outcomes with anti-CTLA-4.

##### T Regulatory (Treg) Subset

Tregs are known for their potent immunosuppressive capabilities that regulate immunological balance [50]. Sakaguchi and his colleagues made the discovery that the transplantation of CD4^+^ T cells might effectively reverse autoimmune disorders in mice that had their thymus removed [51]. These findings suggested that the reduction in suppressive T cells might be a possible factor contributing to autoimmune diseases. These cells in question are often known as Treg and play a vital role in suppressing autoimmune reactions and preserving immunological balance. The predominant subset of Treg CD4^+^ cells has a high expression of the IL-2 receptor (CD25) alpha chain and the FoxP3 transcription factor. These factors are crucial for the production and maintenance of these cells, preventing any harmful effects [52]. Currently, the most extensively studied population of regulatory T cells is the CD4^+^CD25^hi^FoxP3^+^ Treg [53].

Current knowledge distinguishes three Treg subpopulations: naïve or resting Tregs (CD45RA^+^FoxP3^lo^CD25^lo^), correspond to fraction 1 (Fr. I) of the total Tregs; effector Tregs (eTreg) (CD45RA^−^FOXP3^hi^CD25^hi^), fraction 2 (Fr. II) of the total set of Tregs, which come from naïve Treg cells after TCR stimulation and possess a high immunosuppressive capacity; and finally the so-called non-Tregs (CD45RA^−^FOXP3^lo^CD25^lo^), fraction 3 (Fr. III) of the total number of Tregs, which do not possess immunosuppressive capacity but can produce pro-inflammatory cytokines. A significant number of eTreg cells, primarily belonging to the Fr. II subtype, are present in solid human malignancies.

While Treg cells account for just 2–5% of total CD4^+^ T cells in the peripheral blood of healthy individuals, they constitute 10–50% of CD4^+^ T cells in malignancies, with most of them being eTreg cells, as already mentioned. In particular, the presence of Treg cells in the tumor microenvironment (TME) has been linked to decreased survival rates in melanoma patients [54]. Melanoma tumor cells secrete local immune-suppressing chemicals known as transforming growth factor beta (TGF-ß) and IL-10. They facilitate the proliferation of both endogenous Treg cells and the generation of novel-induced Treg cells [55]. Additionally, the Teff/Treg ratio in the TME has consistently been linked to a favorable OS in melanoma patients [56].

Specifically, the presence of Treg cells in the TME of melanoma patients undergoing ICI treatment has been linked to worse results. According to findings from preclinical models, anti-CTLA-4 appears to specifically eliminate intratumoral Treg [57], which consistently expresses CTLA-4 while also activating Teff. The reduction in intratumoral Treg carried by anti-CTLA-4 is achieved by antibody-dependent cell-mediated cytotoxicity (ADCC) in the presence of macrophages expressing FCcRIIIA within the TME [58].

To provide more evidence that anti-CTLA-4 such as ipilimumab facilitates the depletion of Treg cells at the location of the tumor, metastatic biopsies obtained from 11 MM patients treated with ipilimumab showed a direct association between a high ratio of Teff to Treg and the occurrence of tumor necrosis [59]. Also, a comprehensive analysis of biomarkers indicating the likelihood of survival or response to anti-CTLA-4 and anti-PD-1 monotherapy in patients with advanced melanoma revealed a consistent negative correlation between Treg cells and response rates, PFS, and OS in most of the studies [60].

In addition, as most tumor antigens are typical self-antigens, melanoma has the potential to generate tumor-specific Treg cells, which can then inhibit the body’s ability to mount an efficient antitumor response [61]. Comparative in vitro experiments demonstrated that Treg-mediated suppression is more prominent in T cells expressing self-antigenic tumor cells (Melan-A) than in those expressing neoantigens (cytomegalovirus), specifically targeting CD8^+^ T cells. This finding was supported by a study that reported a higher level of Treg-mediated suppression in self-antigen-specific CD8^+^ T cells compared to non-specific CD8^+^ T cells [62]. This might partially elucidate why tumors with a higher number of neoantigens are more vulnerable to anti-PD-1 therapy [63].

However, PD-1 inhibition can stimulate the growth of CD8^+^ T cells and the secretion of cytokines such as IFN-γ and IL-2. These cytokines decrease the number of Treg cells by raising the ratio of Teff cells to Treg cells [63]. Simultaneously, IFN-γ can render Treg cells within tumors vulnerable, making them vulnerable and causing them to lose their capacity to inhibit growth and produce additional IFN-γ while still maintaining their Foxp3 expression and overall characteristics [64].

Administering anti-PD-1 in a mouse model of adenocarcinoma resulted in an increase in the quantity of IFN-γ^+^ Treg cells and a clinical response to anti-PD-1. This finding suggests that the treatment made these cells more susceptible to damage. When Treg cells lacked sensitivity to IFN-γ (using an Ifngr1L/LFoxp3Cre-YFP mouse), rats exhibited total resistance to PD-1 blocking, in contrast to a 40% response observed in WT mice [65]. This finding indicates that the fragility of Treg cells plays a role in determining their responsiveness to immunotherapy.

Experiments conducted on PBMCs derived from 14 individuals with advanced-stage melanoma (stage III or IV) demonstrate that the inhibition of PD-1 with pembrolizumab can enhance the production of cytotoxic T lymphocytes that specifically target melanoma antigens and prevent Treg from limiting the function of these cytotoxic T lymphocytes [66].

In addition, a study using mouse models found that Tregs expressing PD-1 have a greater level of immunosuppressive activity compared to Tregs without PD-1 [67]. Thus, PD-1-blocking antibodies have the potential to stimulate antitumor immunity and restore the functional activity of defective Teff cells by targeting Treg cells.

Finally, in a clinical study exploring this issue, an analysis of 32 individuals with MM treated either with nivolumab or pembrolizumab revealed a notable decrease in the presence of peripheral PD-1^+^ Treg cells (from 23% to 8.6%, *p* < 0.05) following the initial cycle of immunotherapy. This reduction was observed specifically in responders to treatment [68] and correlated to PFS.

Although higher Treg presence in melanoma patients is generally associated with a poor prognosis, the effect of ICI treatment on this CD4^+^ subtype appears to differ between anti-CTLA-4 and anti-PD-1. While anti-CTLA-4 directly suppresses Treg through an ADCC mechanism, anti-PD-1 promotes CD8^+^ Tc expansion and consequently increases the Teff/Treg ratio. In addition, anti-PD-1 targets a subtype within Treg with a more fragile phenotype, identified as PD-1^+^ Treg and IFN-γ^+^ Treg, which prevents its suppressive effect on CD8^+^ Tc activation.

#### 2.1.3. CD8^+^ T Cells

CD8^+^ T cells, unlike CD4^+^ T cells, serve as cytotoxic T lymphocytes (Tc) and can directly eliminate tumoral and infected cells [36]. They possess the ability to identify antigens presented by MHCI, a molecule present in nearly all cells within the body. Subsequently, they can induce cell death by secreting chemicals that degrade cells, such as perforin and granzymes, or by initiating apoptosis via Fas.

Multiple investigations have demonstrated the essential role of CD8^+^ T cells in the immunological response to MM [69]. A higher intratumoral Teff to Treg ratio, as indicated before [56], has been linked to a higher ORR and OS in patients with MM who received anti-CTLA-4. Tumeh et al. [70] examined samples obtained from 46 individuals with advanced melanoma both before and after receiving pembrolizumab therapy. The researchers discovered that the quantity of CD8^+^ T lymphocytes located near the invasive margin and intratumoral was the most accurate predictor of response to treatment [70].

An alternative investigation examined PBMC samples from 131 patients with MM who received treatment with either anti-PD-1 alone or in combination with anti-CTLA-4 [71]. On the 21st day, the patients who responded to treatment showed a higher number of big CD8^+^ T clones (those that make up more than 0.5%) in their PB compared to both healthy individuals and non-responsive patients. Flow cytometry analysis demonstrated a robust association between the abundance of big clones and the quantity of Tem CD8^+^ cells in PBMCs collected prior to and after treatment. In a separate group of MM patients who received ICI studied within the same work, scRNA-seq was performed to investigate the extent to which transcriptional alterations occur in CD8^+^ T cells based on their phenotypic subgroup and clonal size. The researchers discovered that the Tc effector subset had the highest number of genes expressed differently and maintained a consistent clonal size following pembrolizumab (*n* = 4) and nivolumab plus ipilimumab treatment (*n* = 4).

For instance, the activation of the IFN-γ pathway [72] demonstrates that the CD8^+^ T cell response to immune checkpoint inhibitors involves several components, such as cell division during mitosis and enhanced cytotoxicity.

Furthermore, a distinct investigation was carried out on a group of 31 individuals with advanced melanoma who had treatment with ipilimumab and stereotactic ablative radiation therapy (SBRT). The study revealed that patients who obtained therapeutic benefit, namely CR, PR, or SD ≥6 months, exhibited a higher prevalence of peripheral T cells CD8^+^. Furthermore, these individuals had an elevated CD8^+^/CD4^+^ cell ratio and a greater proportion of CD8^+^ cells expressing PD-1 [73], further described in Section 2.1.4.

Eventually, it is crucial to consider the precise location of the metastases in the tissue samples that were examined in each study. To address this issue, research was conducted on patients with MM who were administered ipilimumab. The study examined immune subsets in lymph node tissue samples, as well as metastases located on the surface and beneath the skin [74]. The median densities of immune cells in lymph node metastases were consistently twice as high as those in cutaneous or subcutaneous metastases for most of the indicators examined. IHC analysis revealed that those who responded to treatment had a higher abundance of CD8^+^ T cells compared to those who did not react. This difference was observed specifically in lymph node metastases [74].

To date, there is strong evidence that CD8^+^ T cells confer a better prognosis in melanoma patients and that their expansion occurs after anti-CTLA-4 and anti-PD-1 monotherapy as well as anti-PD-1 plus anti-CTLA-4 combination treatment. Furthermore, a higher intratumoral or PB frequency seems to correlate with a better outcome, regardless of the ICI strategy.

#### 2.1.4. Dysfunctional Programs: From T Progenitor Exhausted (Tex Prog) to Terminally Exhausted (Tex Term) and Reinvigoration

T cells go through different phases of maturation from Tn to Tcm, Tem, and Teff, until reaching a final phase of exhaustion that takes place in the face of prolonged chronic activation and is called exhausted T cells, characterized by deficient secretion of cytokines and expression of co-inhibitory signals such as PD-1 and TIM-3 [75]. These two markers, which are immune checkpoints considered co-inhibitory signals, make it possible to identify CD8^+^ T cells at opposite ends of the functional spectrum. It is considered that CD8^+^TIM-3^−^PD-1^−^ cells are highly functional with a great effector capacity of the immune response, while CD8^+^TIM-3^+^PD-1^+^ cells are highly dysfunctional, and each population has its own transcriptional profile [75]. A single-cell analysis of 25 immune infiltrates from melanoma tumors revealed a separation between bystander Tc and a population that shows a continuous progression from a transitional early effector to a dysfunctional T cell state [76]. This suggested differentiation pathway seems to be something that occurs in all CD8^+^ T cells in melanoma, according to several studies [76].

Within the exhausted CD8^+^ TILs, there are two distinct subpopulations: Tex prog, which preserves their ability to perform several functions and survive over a long period of time, and Tex term [77]. CD8^+^ Tex prog cells show a middle level of PD-1 expression and are exposed to the CXCR5 chemokine receptor. Conversely, CD8^+^ Tex terms have elevated expression of PD-1, TIM-3, and other co-inhibitory receptors [78].

There are notable functional distinctions between these two subpopulations. Researchers have discovered that Tex cells that expand after anti-PD-1 treatment display a Tex prog phenotype [79]. In a study that analyzed CD8^+^ TILs from baseline tumor samples of 25 melanoma patients treated with nivolumab plus ipilimumab, researchers identified TCF1^+^PD-1^+^CD8^+^ T cells consistent with the Tex prog phenotype in almost all biopsies [79]. No significant differences in the frequency of TCF1^+^PD-1^+^CD8^+^ T cells were observed between responders and non-responders. However, in responders, on the other hand, the duration of the response was linked to the number of TCF1^+^PD-1^+^CD8^+^ T cells, and a higher number of these CD8^+^ T cells was significantly associated with longer PFS and OS (*p* < 0.05) [79].

Another study of neoadjuvant anti-PD-1 in 20 locally advanced melanoma patients showed a rapid pathological response associated with the accumulation of CD8^+^ Tex cells in the tumor at 3 weeks and reinvigoration in the blood as early as 1 week, as determined by the expansion of Ki67^+^PD-1^+^ and Ki67^+^PD-1^+^CTLA-4^+^CD8^+^ T cells [80]. Most of the CD8^+^ T cells in the tumor were CD45RA^lo^CD27^hi^ and expressed PD-1, TIM-3, CTLA-4, LAG-3, TIGIT, and CD39 [80]. Also, a lot of these CD8^+^ T cells had high Eomes levels but low T-bet levels, which is in line with the Tex phenotype [81] (further explained in Section 2.1.5). In the same study, a high percentage of Eomes^hi^T-bet^lo^ Tex was associated with clinical benefit, while Treg proliferation was associated with relapse and poor disease-free survival (DFS).

Consistently, in a systematic Medline search for biomarkers of survival or response to ICI in melanoma patients [60], PD-1^+^CD8^+^ T cells were the only biomarker significantly associated with OS in more than one study [60]. In accordance with these findings, in two studies of a cohort of 20 advanced MM patients each, samples from patients treated with pembrolizumab or nivolumab [82,83], PFS, and response were positively correlated with ≥20% higher proportions of CD8^+^ T cells in tumor biopsies characterized by high surface expression of CTLA-4 and PD-1 [82,83], regardless of the anatomical site of biopsy or prior therapy.

Further evidence was provided by another study [84] that used PBMCs of 29 MM patients before and after pembrolizumab treatment to identify pharmacodynamic changes in circulating exhausted CD8^+^ T cells. However, clinical failure in many patients was not due to a failure to induce immune reconstitution but to an imbalance between T cell reconstitution and tumor burden. This conclusion was reached by finding that a Ki67 to tumor burden ratio ≥1.94 at 6 weeks was associated with a better outcome based on ORR, PFS, and OS. An increase in Ki67 expression was observed in CD8^+^ T cells among melanoma patients (*p* < 0.05) who received anti-PD-1 treatment, mostly in the PD-1^+^CD8^+^T subgroup (*p* < 0.05), indicating the presence of an immune response before treatment [84]. The majority of the Ki67^+^CD8^+^ T cell population from patients that responded had a phenotype of CD45RA^lo^CD27^hi^, as well as cells with elevated levels of CTLA-4 and PD-1. Furthermore, the upregulation of Ki67 expression was more prominent in PD-1^+^CD8^+^ cells compared to PD-1^−^CD8^+^ cells (*p* < 0.05). At baseline, about half of the PD-1^+^CTLA-4^+^ cells had Ki67 on them. After the therapy, that number rose to about 75%, which is in line with the recovery of Tex cells [84].

All these data show that in melanoma patients, blocking the inhibitory immune checkpoint PD-1, which consequently prevents binding with its ligand, PDL1, triggers the regeneration of T cells within the tumor. This promotes the transformation of Tex into Teff cells, which contribute to the antitumor response. Viral infections, as mentioned above, have described a hierarchy of CD8^+^ T cell dysfunction [83]. Partially exhausted CD8^+^ T cells have been defined as cells that can produce IFN-γ but lack the ability to produce TNF-α and IL-2, a phenotype consistent with tumor-infiltrating CTLA-4^hi^PD-1^hi^CD8^+^ T cells. In two cohorts of patients with advanced melanoma treated with ICI, anti-PD-1, and anti-TIM-3, there was a strong correlation between the proportions of PD-1^hi^CTLA-4^hi^ cells within the tumor-infiltrating CD8^+^ T cell subset and the ORR and PFS [85]. The ORR was 0% (0 of 6) in the low CTLA-4^hi^PD-1^hi^ T cell expression group, compared to 85.7% in the higher frequency CTLA-4^hi^PD-1^hi^ T cell expression group [85].

Finally, researchers from this same work [85] used a preclinical model to find out how many PD-1^−^CD8^+^ or PD-1^+^CD8^+^ TILs were present after each cycle of anti-TIM-3 and anti-PD-1 therapy [85]. After three cycles, the number of PD-1^−^CD8^+^ TILs increased, whereas the number of PD-1^+^CD8^+^ TILs did not change [85]. They hypothesize that PD-1^−^CD8^+^ TILs are in a less advanced state of differentiation than PD-1^+^CD8^+^ TILs. Therefore, they anticipated that PD-1^−^CD8^+^ TILs would have a greater capacity for revitalization and the ability to sustain long-term immunity. This study [85] classified T cells based on their memory state and the expression of three markers: CX3CR1, CD62L, and Slam7. They identified three subsets within the TILs PD-1^−^CD8^+^ group: the naive subset (CD62L^hi^SLAMF7^hi^CX3CR1^−^PD-1^−^CD8^+^), the memory precursor phenotype subset (CD62L^−^SLAMF7^hi^CX3CR1^−^PD-1^−^CD8^+^), and the effector phenotype subgroup (CD62L^−^SLAMF7^hi^CX3CR1^+^PD-1^−^CD8^+^). Interestingly, memory precursor phenotype and effector phenotype were the subpopulations that expanded more upon the ICI treatment applied.

Therefore, as previously stated [34], CX3CR1 has been identified as an indicator of T cell differentiation within the memory compartment and a prognostic factor for the effectiveness of ICI treatment. The data collected in this work [85] demonstrated that the transcription alterations caused by the inhibition of immune checkpoints corresponded with the development of memory and effector capabilities in CD8^+^ TILs. This phenomenon was observed to a greater extent in the PD-1^−^CD8^+^ subpopulation compared to the PD-1^+^CD8^+^ subpopulation.

Consequently, taking all these findings into account, T cells that are more prompt to experience reinvigoration driven by anti-PD-1 monotherapy or in combination with anti-CTLA-4 treatment are the ones that are less differentiated; however, it can be a balance between the differentiation within exhaustion and memory states, which defines the phenotype that expands more frequently after anti-PD-1-based treatment.

#### 2.1.5. Transcription Factors and Encoding Gene Regulators of Lymphocytes T Cell Populations

##### Tcf7 and Tcf1/TCF1

Tcf7 is a gene that encodes for a transcription factor called Tcf1/TCF1. The primary function of this is to sustain the survival of CD8^+^ memory T cells and facilitate their transformation into effector T cells [86]. TCF1 is essential for differentiation within the Th1/2/17/22 cell lineage. Additionally, it plays a crucial role in controlling the suppressive function of Treg in the immune system [87].

As previously mentioned, it has been shown that Tex prog TILs, which were developed following the anti-PD-1, had significant stem-like characteristics, including the capacity for self-renewal and the potential to transform into Tex terms. TILs that express TIM-3, known as Tex prog TILs, have lower levels of the co-inhibitory receptors PD-1 and LAG-3 compared to TILs that express both TIM-3 and PD-1, considered Tex term. Significantly, Tex prog TILs specifically correlate with the presence of TCF1 expression, whereas Tex term TILs lack this expression [87].

In preclinical research, a mouse model with a deletion of Tcf7 (E8i-Cre^+^Tcf7fl/fl) in mature CD8^+^ T cells was used [87]. The PD-1^−^CD8^+^ TIL exhibited a significant reduction in the number of memory precursor-like subsets in this animal. These findings indicate that Tcf7 is crucial for the formation and preservation of the memory precursor-like subgroup. Also, in a study conducted on 48 patients with advanced melanoma, 35 patients received anti-PD-1 monotherapy, 11 received anti-PD-1 and anti-CTLA-4, and 2 received anti-CTLA-4 monotherapy [88]. Among all patients, 31 responded to the treatment, while 17 did not. Immune cells from tumor samples were chosen based on the CD45^+^ marker using CellProfiler 2.2.0 [83], an automated imaging system, and scRNA-seq was performed. The scRNA-seq analysis revealed that responders had a >1 ratio of TCD8^+^TCF7^+^ cells to TCD8^+^TCF7^−^ cells. Based on this discovery, the study determined that a higher proportion of Tex prog cells compared to Tex term cells is associated with a positive response to anti-PD-1-based therapy in patients with MM [88].

Siddiqui et al. conducted research where they discovered the presence of TCF1^+^PD1^+^CD8^+^ T cells within tumors in animal models following vaccination and checkpoint blockade treatment. These cells had characteristics like stem cells and played a crucial role in tumor control [89]. The previous understanding was that depletion impeded the formation of memory T cells. However, the identification of a specific group of virus-specific CD8^+^ T cells, which express both PD-1 and Tcf1 and are able to maintain the immune response during chronic infection, has revealed a link between T cell memory and exhaustion [89]. Therefore, Tcf1 is crucial in the development and functioning of CD8^+^ Tcm cells following the resolution of an acute infection [89].

Therefore, there is a link between the exhaustion phase and the differentiation memory phase, which defines the phenotype that expands more frequently after ICI treatment, and TCF1 has a crucial role in determining which differentiation states are found in T cells within memory and exhaustion subsets.

##### Eomes and T-Bet

Regulatory transcription factors, such as Tcf1 (encoded by Tcf7), already reviewed, Eomes, and T-bet, play a crucial role in governing the functional development of memory T cell subsets and their transformation into effector cells. Furthermore, T-bet and Eomes promote Th1 effector activity in cytotoxic T cells by increasing the expression of IFN-γ and cytotoxic granules, such as granzyme B (GzB). Afterward, the absence of T-bet leads to impaired function of effector T cells [90].

T-bet and Eomes work together to sustain the antiviral CD8^+^ T cell population during a long-lasting viral infection. T-bet^hi^ cells exhibit a low inherent rate of replacement, but they undergo cell division in the presence of persistent antigens, resulting in the production of Eomes^hi^ terminal progenitors. Modified genetic elimination of either subset resulted in failure to control chronic infection, which suggests that an imbalance in differentiation and renewal could underlie the collapse of immunity in humans with chronic infections [85,86,91].

A preclinical investigation indicated that inhibition of the PD-1 pathway may result in T-bet collaborating with Eomes to bind to the Pdcd1 gene or competing with it for binding. This competition leads to increased transcriptional repression. This interaction among T-box transcription factors is probable to happen in additional genes. Eomes can partially counteract the function of T-bet and presumably adjust the program of Tex by controlling its location inside the nucleus and potentially competing for DNA binding [92].

Elevated Eomes expression is necessary for the development of long-lasting memory cytotoxic T cells. If Eomes is absent, it leads to impaired maintenance of memory populations [92]. In addition, CTLA-4 decreases the transcriptional expression of Eomes, resulting in a reduced number of CD8^+^ T cells that generate IFN-γ and GzB. Wei et al. [41] discovered a group of Eomes^hi^T-bet^hi^CD8^+^ T cells that were partly exhausted and showed considerable expansion when treated with anti-CTLA-4. In addition, Gide et al. demonstrated that in a group of 158 melanoma patients who received treatment with either anti-PD-1 alone or plus anti-CTLA-4, the presence of Eomes^+^CD69^+^CD45RO^+^ Tem cells in tumor samples was linked to a response and longer PFS [93].

Eomes ultimately facilitates the development of the Tex term subpopulation [84], whereas T-bet and Tcf7 facilitate the development of the Tex prog exhausted subpopulation [79]. As previously stated, following anti-PD-1 administration, the Tex prog population undergoes expansion and originates from the subgroup of cells that are in a state of terminal exhaustion [79]. However, they must evolve to a terminal exhaustion state to promote a long-lasting response, and at these two points, T-bet and Eomes intervene.

Again, we can see the link between memory differentiation states from more precursor subsets to effector cells induced by Eomes expression that also facilitates the transition within exhausted population phases from progenitor to terminal exhausted cells.

#### 2.1.6. Double Negative T Cells (DNTs)

Within T cells, defined by the expression of CD3^+^, apart from the two major groups, CD4^+^ and CD8^+^, a subset of cells is gaining interest that is characterized by the expression of TCR αβ or γδ but in the absence of surface T markers such as CD4, CD8, and CD56. Based on this, these are called double-negative T cells (DNTs) because they do not express any of the markers that classify T cells into large groups [94]. They represent about 3% of total circulating T cells and have been shown to play an important role in preventing organ-versus-host disease [95] and in modulating the magnitude of the immune response [96]. In one study, including 68 patients with advanced melanoma treated with anti-CTLA-4 or anti-PD-1, the median level of circulating DNT decreased, while CD4^+^ and NK cells increased in patients who responded to treatment compared to those who did not [97].

So far, no further clinical data about DNT’s potential predictive response or prognostic role in melanoma patients treated with ICI have been published. According to these findings, DNTs could hinder immune responses mediated by ICI, but additional research is needed to confirm this assumption.

### 2.2. Chemokines and Outcomes of Melanoma Patients Treated with ICI

#### 2.2.1. CXCR3

CXCR3 is a chemokine receptor that promotes the migration of Th1 cells and Teff CD8^+^ cells into inflamed tissues, including tumors [98]. CXCR3 is mostly expressed in activated T cells and NK cells that release IFN-γ [98]. CXCL9 and CXCL10 act as ligands for CXCR3 and are generated in response to IFN-γ activation. Moreover, the IFN-γ-CXCL9/10-CXCR3 pathway serves a vital role in recruiting IFN-producing cells [98].

A study of 43 melanoma patients receiving pembrolizumab therapy found a significant increase in the percentage of CXCR3^+^ T cells in the blood following the first infusion [98]. However, the rate of expansion decreased following the second infusion in patients who responded well, whereas individuals with progressive disease continued to routinely show large percentages of these cells. The presence of CXCR3^+^ T cells in the circulation suggests a deficiency of IFN-γ-producing T cells stepping into tumors, leading to an insufficient response to anti-PD-1 treatment [98]. Consistently, individuals who did not respond exhibited a significant decrease in their levels of CXCL9/10. Furthermore, the study found that when anti-CXCR3 antibodies were used in preclinical experimental settings, there was a consistent acceleration of tumor growth in mice that had B16-F10 tumor cells [98]. In addition, the concurrent use of intramuscular CXCL9/10 and intraperitoneal anti-PD-1 administration successfully inhibited tumor growth. Therefore, this study hypothesized that the increase in CXCR3^+^ T cells in the blood of treatment-resistant patients might be caused by a decrease in the activation of CXCL9 and CXCL10 inside the tumor microenvironment.

Further, in this setting, flow cytometry analysis to investigate the presence of circulating or CD8^+^ TILs in 52 patients with MM found a direct relationship between the expression of CXCR3 in CD8^+^ TILs and prolonged OS [99]. Also, in another scientific study that analyzed plasma samples from 28 patients with advanced melanoma who were treated with pembrolizumab and nivolumab plus ipilimumab, responders (*n* = 18) had a greater quantity of CXCL9 and CXCL10 compared to non-responders (*n* = 10) [100].

Furthermore, separate research examined the immune profile and transcriptomics of 158 tumor samples obtained from advanced melanoma patients who received either anti-PD-1 monotherapy (*n* = 63) or a combination of anti-PD-1 and anti-CTLA-4 (*n* = 57) [101,102]. CXCL9 and CXCL10 were identified as two of the most prevalent and significantly increased chemokines following the administration of a combined immune checkpoint blockade (*p* < 0.005). Significantly, it was shown that the expression of CXCL9 and CXCL10 was notably greater in tumors obtained from individuals who responded to treatment, both before and throughout therapy. Moreover, in these individuals, increased levels of CXCL9 and CXCL10 were linked to improved OS and a greater presence of CD8^+^ T cells within the tumor [101,102].

More recently published research has shown that the CXCL10/CXCR3 axis has a paracrine function in the antitumor activities of Th1 cells, Tc, and NK cells. However, CXCL10/CXCR3 signaling promotes tumor cell proliferation, angiogenesis, and metastasis when mediated by an autocrine pathway in tumor cells. In general, the specific kinds of receptors and cells involved appear to be the determining factors in the contradictory effects of CXCL10 on tumor growth. However, further research is needed to fully understand the underlying processes [103].

Preclinical evidence demonstrates that the injection of B16F10 tumor cells with decreased CXCR3 expression into C57BL/6 mice greatly decreases the occurrence of metastasis in lymph nodes [104]. Thus, administering full Freund’s adjuvant to the mice prior to therapy resulted in an increase in the levels of CXCL9 and CXCL10 in the draining lymph nodes. As a result, there was a significant 2.5–3.0-fold increase (*p* < 0.05) in the number of B16F10 cells that spread to lymph nodes, forming bigger clusters. Significantly, the growth of metastasis was greatly inhibited by lowering the expression of CXCR3 in B16F10 cells using reverse RNA or by dosing mice with antibodies against CXCL9 and CXCL10 [104].

Considering all published data to date, the IFN-γ-CXCL9/10-CXCR3 pathway could have an antitumoral role when driven by paracrine signals and a protumoral role when driven by autocrine signals. This second occasion is caused by tumor cells.

#### 2.2.2. IL-8

CXCL8, commonly referred to as IL-8, is a cytokine that exhibits several actions, including immunological effects like attracting neutrophils and promoting angiogenesis. The alpha and beta receptors, CXCR1 and CXCR2, are G-protein-coupled receptors. Various cell types, such as monocytes, neutrophils, epithelial, fibroblast, endothelial, mesothelial, and cancer cells, release IL-8 when exposed to inflammatory stimulation. It has a crucial function in the process of inflammation and the healing of wounds. Additionally, it can attract T cells and non-specific inflammatory cells to areas of inflammation by activating neutrophils. Patients diagnosed with various types of cancer have increased levels of IL-8, which seem to be associated with a more advanced tumor stage, grade, and disease severity [105].

Tumor-derived IL-8 has the capacity to have significant impacts on the tumor microenvironment [106]. It facilitates the maintenance of cancer cells in the epithelial-mesenchymal transition (EMT) state, enabling their mobility and invasiveness. Additionally, it promotes angiogenesis [107]. An instance of this is when cancer cells release IL-8, which can stimulate the growth and survival of cancer cells themselves through autocrine signaling pathways [106]. The attraction of myeloid-derived suppressor cells due to IL-8 secretion, in turn, hinders the infiltration of T cells and weakens the expression of IFN-γ. Quantifying the amounts of IL-8 in serum is a straightforward process that may be easily conducted using ordinary blood samples in regular clinical practice [105,108].

On this topic, a study was conducted with 1344 patients who had advanced solid tumors treated with nivolumab and/or ipilimumab, everolimus, or docetaxel within clinical trials [105]. This study found that elevated levels of IL-8 in the bloodstream, specifically ≥23 pg/mL, were linked to poorer OS, independently of tumor type. Notably, the subgroup of patients with the strongest correlation between IL-8 levels and prognosis were MM patients treated with nivolumab plus ipilimumab within the CheckMate 067 trial.

Moreover, Schalper et al. conducted an analysis of data from four clinical trials that included patients with melanoma, among other solid tumors, who were treated with ICI [109]. They found that greater baseline levels of IL-8 were consistently associated with poorer survival outcomes in patients who received anti-PD-1, anti-CTLA-4, or a combination of both treatments. Furthermore, patients with elevated IL-8 exhibited a decrease in ORR. The levels of IL-8 in circulation showed a favorable correlation with the expression of the CXCL8 gene inside the tumor and the number of neutrophils and monocytes in the peripheral blood.

To provide more evidence on this matter, we reference another study that examined the serum level of IL-8 in 29 MM patients who were treated with either nivolumab or pembrolizumab as a single therapy or nivolumab in combination with ipilimumab. Levels were measured at the beginning of the treatment and again 2–4 weeks later [110]. Among patients who showed a response to treatment, the median levels of IL-8 in their blood decreased dramatically at the time when they had the best response, compared to the initial levels. Early changes in serum IL-8 levels were also strongly linked to responses to anti-PD-1 and anti-PD-1 plus anti-CTLA-4, but not baseline levels. Also, patients who were treated with anti-PD-1 and whose serum IL-8 levels dropped early had a significantly longer OS than patients whose levels rose early (*p* < 0.05).

Finally, in a multitumor study, serum IL-8 levels correlated positively with tumor burden and negatively with OS and ORR in patients with various solid cancers, including MM patients (*n* = 24) treated with BRAFi and ipilimumab [111].

However, all these studies were published several years ago, and the potential efficacy of anti-IL-8 is being evaluated within clinical trials in the early development phases.

### 2.3. Distinct Effects of Anti-CTLA-4 and Anti-PD-1 on Lymphocyte Populations

Immune checkpoint blockade with anti-CTLA-4, anti-PD-1 monotherapy, or combined therapy results in a distinct signature of gene expression changes in T cells in vivo, as previously exemplified [41]. Both CD4^+^ and CD8^+^ T cells induced Ki67 in a flow cytometric analysis of pre- and post-therapy PBMC samples from 48 MM patients and mouse melanoma models treated with CTLA-4 and PD-1 blockade combinations and monotherapy [49], with the Ki67^+^ cells exhibiting a CD45RO^+^ memory phenotype. However, combination therapy increased GzB, but neither anti-CTLA-4 nor anti-PD-1 monotherapy did. Anti-PD-1 alone led to the expansion of certain CD8^+^ T cell subsets that entered tumors and were worn out. In contrast, anti-CTLA-4 induces the expansion of an ICOS^+^T-bet^+^ Th1-like CD4^+^ effector population, as well as specific subsets of Tex CD8^+^ T cells in this study [49].

Another study, already mentioned, conducted by Wei et al., used a mass cytometry-based approach to analyze PBMCs from 48 patients treated with either ipilimumab monotherapy (*n* = 13), pembrolizumab monotherapy (*n* = 22), or nivolumab plus ipilimumab +combination therapy (*n* = 13). Immune profiling and analyzing T-cell responses revealed that terminally differentiated CD8^+^ effector T cells expanded only after combination therapy, and combination therapy increased Th1-like CD4 effector T cells greater than anti-CTLA-4 alone, but anti-PD-1 monotherapy failed to do so [41].

### 2.4. Impact of Anti-PD-1 and Anti-LAG-3 Combination Upon Lymphocyte Populations

To date, there is a paucity of data analyzing the impact of the anti-PD-1 plus anti-LAG-3 combination on lymphocyte T populations and their potential role as predictive biomarkers of response to this particular ICI regimen. An animal model consisting of tumor-bearing Msh2loxP/loxP;TgTg (Vil1-cre) mice treated with an anti-PD-1 and anti-LAG-3 combination showed reduced circulating Tregs and lower levels of Tex cells positive for CTLA-4, LAG-3, and PD-1 [112]. Higher expression of Tcf1 was also observed with both ICI treatments. In the peripheral blood, however, increased numbers of Th and Tc cells were observed with anti-PD-1 treatment and slightly decreased numbers with anti-LAG-3 treatment. Moreover, the effects of the two ICIs on Tex were different. Anti-PD-1 treatment reduced the number of CTLA-4^+^ T cells and LAG-3^+^ T cells [112]. In contrast, the effect of the anti-LAG-3 antibody on exhaustion markers was more general, resulting in a decrease in all T cell populations studied, especially PD-1^+^ T cells. Within the memory phenotype compartment, CD4^+^ Tcm and Tem were more expanded with anti-LAG-3 than anti-PD-1 treatment, but CD8^+^ Tn were decreased. Ki-67 expression was analyzed on TILs; higher expression was observed after ICI treatment, and these effects were more pronounced after anti-PD-1 than anti-LAG-3 [112].

In a phase II trial evaluating nivolumab and relatlimab as neoadjuvant therapy in patients with stage IIIB to IV melanoma, TIL subsets were analyzed by mass and flow cytometry [113]. Unsupervised analysis identified a subset of effector CD8^+^ T cells and a subset of memory CD4^+^ T cells, which were increased in tumor samples obtained after treatment in responders compared to non-responders. In peripheral blood, a trend toward increased CD8^+^ EOMES^+^ T cells was observed in responders compared to non-responders after treatment, with the greatest differences seen at week 5 since treatment started. However, the study emphasizes that the low number of optimal samples in non-responders may have partly conditioned these results.

## 3. Discussion

Although the therapeutic landscape for advanced melanoma has radically changed in the last decade since the approval of the different ICI regimens and BRAF/MEKi, a significant percentage of patients still do not benefit from these therapies [1,2,3,4,5,6,7,8,9,10,11,12]. The aim of this review is to evaluate the potential role of lymphocyte T subsets as prognostic or predictive biomarkers for ICI treatment in melanoma patients, regardless of mutational tumor landscape, by reviewing all meaningful published data on this topic. Some interesting and possibly useful results were found. Notwithstanding, it is still necessary to determine a cut-off value for the percentage distribution of lymphocyte subsets in the PB, the appropriate timepoint for their determination (at baseline and/or longitudinally during treatment), and whether the PB distribution matches the TME distribution.

First, considering the published data, we believe that memory T cells and exhausted T cells are crucial in determining the efficacy of ICI, particularly anti-PD-1-based treatment. Although there are conflicting results regarding the role of CD4^+^ memory T cells, more solid evidence supports the association of higher frequency in PB and TILS of CD8^+^ Tcm and Tem cells with better response to anti-PD-1 therapy and longer duration of this, contributing to longer PFS and OS [26,33]. In comparison, higher levels of Tn in both compartments, CD4^+^ and CD8^+^, generally correlate with poorer outcomes. We also highlight the chemokine CX3CR1 as an important indicator of T cell differentiation within the memory subset, and its higher expression in CD8^+^ T cells may help predict a better response to anti-PD-1 [34].

As a natural process following chronic activation of the immune response, there is a progressive maturation of T cells from Tn cells to Tcm and Tem to Teff cells and finally, Tex, which show features of dysfunction such as impaired cytokine secretion and expression of markers of exhaustion such as PD-1 and TIM-3 [76,77,78]. Thus, analyzing both markers’ expression enables the identification of CD8^+^ T cells at opposite ends of the exhaustion spectrum [75].

Moreover, it is also relevant to mention the T exhaustion process from progenitor Tex cells to Tex term and the fact that anti-PD-1-based treatments have been shown to reinvigorate these cells, in particular CD8^+^ T cells expressing PD-1, making them differentiate, proliferate, and become re-activated, being able to mediate an antitumor response in PB and intratumorally [80,83]. In this direction, the higher presence of CTLA-4^hi^PD-1^hi^CD8^+^ T cells in peripheral blood or TILS has been correlated with a good prognosis and ICI response, especially anti-PD-1-based treatments, in several studies [80,83,85].

Several transcription factors are involved in the regulation of this process. Eomes facilitate the development of Tex term cells, whereas T-bet and TCF7 are associated with Tex prog cells. Anti-PD-1 therapy primarily expands Tex prog cells [79], which are derived from cells in a state of terminal exhaustion. CD8^+^ T cells expressing TCF7 and its transcription factor Tcf1/TCF1 are more frequently expanded after ICI treatment in responders [87,88,89].

However, in a study [85] using a preclinical model, they observed that after three cycles of anti-TIM-3 and anti-PD-1 therapy, the number of PD-1^−^CD8^+^ TILs increased while the number of PD-1^+^CD8^+^ TILs did not change. They hypothesize that PD-1^−^CD8^+^ TILs are in a less advanced state of differentiation than PD-1^+^CD8^+^ TILs. In the same study, they identified three subsets within the PD-1^−^CD8^+^ group of TILs and found that memory precursor and effector phenotypes were the ones that expanded more after ICI treatment. One of the key markers to subclassify these subsets was CX3CR1.

Therefore, taking all these findings into account, T cells that are more responsive to ICI treatment are the ones that are less differentiated; however, it may be a balance between the exhaustion phase and the differentiation phase that defines the phenotype that expands more frequently after ICI treatment.

Also, CD8^+^ T cells, in general, but especially those with a cytotoxic phenotype, have been widely associated with a good prognosis in melanoma patients, and a high infiltration of CD8^+^ T cells in tumors has been considered a strong predictor of response to ICI [69,70,71]. Furthermore, the presence and expansion of large CD8^+^ T cell clones are critical for anti-PD-1-mediated responses. A higher CD8/CD4 ratio is also predictive of better outcomes with ICI treatment, presumably at the expense of the CD8^+^ cytotoxic subset, as these express IFN-γ, a key element for successful antitumor immune responses [73].

In contrast, the literature is inconclusive in clearly linking an increase in Th1, Th2, or Th17 cells after ICI treatment to improved responses. However, combined blockade with anti-PD-1 and anti-CTLA-4 may potentiate Th1 cells, which are characterized by their production of IFN-γ and have been associated with a better prognosis in melanoma patients [41,49].

However, Treg, especially eTregs (with greater suppressive capacity), which have been extensively correlated with poor prognosis in several tumors, including melanoma [59], have been shown to hinder responses mediated by anti-PD-1 therapy and anti-CTLA-4 [57,58,59,60,61,62,63,64,65,66]. PD-1^+^ Tregs may be particularly affected by anti-PD-1 blockade, and their reduction correlates with better responses [86]. The reduction in intratumoral Treg by anti-CTLA-4 is achieved by ADCC in the presence of FCcRIIIA-expressing macrophages in the TME [58], whereas anti-PD-1 agents may have a more potent indirect mechanism by enhancing Teff, thereby increasing the Teff/Treg ratio. Thus, a lower ratio of Teff/Treg has been associated with resistance to ICI.

Data on DNTs are limited for the scenario of this review, but evidence suggests that they may play a suppressive role in antitumor immunity. Further research is needed to understand their exact function and potential impact on immunotherapy outcomes [94,97].

However, it should be taken into consideration that the differences in lymphocyte T population impact may depend on the treatment applied in some situations. For instance, we have mentioned differences between anti-PD-1 and anti-CTLA-4 monotherapy on Th1 cells in comparison to the anti-PD-1 and anti-CTLA-4 combination. Also, other ICI combinations may induce different changes, such as anti-PD-1 plus anti-LAG-3; preliminary data show that anti-LAG-3 produces a more general expansion of Tex compartment ant Tcm and Tem in contrast to anti-PD-1, which enhances more T cell proliferation.

Importantly, the representation of lymphocyte populations in PB may or may not reflect the TME, depending on the lymphocyte subset. Some subsets, such as Trm, may also be important for a successful immune response after ICI, but they are not represented in PB, which is less accessible for routine determination [40,42].

Finally, the measurement of some circulating cytokines may also be relevant. For instance, the IFN-γ-CXCL9/10-CXCR3 pathway can play both antitumor and protumor roles depending on the signaling context [98,99,100,101,102,103,104]. When driven by paracrine signals, this pathway can enhance the immune response against tumors by promoting the activity of Th1, Tc, and NK cells and guiding them to the TME. Conversely, when driven by autocrine signals from tumor cells, this pathway can facilitate tumor growth and metastasis. Understanding these dual roles and analyzing which one is acting in each situation is crucial for carefully evaluating the potential use of CXCR3/CXCL9/CXCL10 plasma determination as a predictive biomarker for identifying ICI responders among MM patients.

Another protumoral cytokine that can be easily determined in PB is IL-8 [105,111]. Several data have shown their correlation with poor prognosis in a variety of solid tumors, including melanoma, and it could help identify non-responders to ICI [108,109,110,111]. Nevertheless, as mentioned above, most of the studies were published some years ago, and, to our knowledge, clinical trials exploring the potential efficacy of anti-IL-8 are still in their early development phases.

**Table 1 ijms-25-09506-t001:** Lymphocyte T subsets and findings in melanoma patients treated with ICI.

Lymphocyte T Cell Population		Patients Population	Observation	Ref.
Memory populations	Tn CD4^+^	Resectable stage III-IV melanoma treated with neoadjuvant Ipi	↓ after treatment	[26]
	Tn CD8^+^	MM was treated with Pem	↓ in responders	[31]
	Tcm CD8^+^		↑ in responders	
	Tcm CD8^+^	MM was treated with Ipi	≥13% Tcm CD8^+^ in long responders	[29]
	Tcm	MM was treated with Ipi	↑ in pts with DC	[28]
	Tem			
	Tem CD8^+^	MM treated with Nivo or Pem or Nivo+Ipi	↑ after 3 weeks of responders	[33]
	Tem CD4^+^	MM treated with Nivo+Ipi	↑ in better PFS	[30]
		MM was treated with Pem	↓ in responders	[31]
	Tcm/Teff	MM and NSCLC treated with Nivo	↑ better PFS	[35]
	Trm CD8^+^	MM treated with anti-PD-1	↑ in responders	[37]
		MM treated with anti-PD-1	↑ better melanoma-specific survival and ORR	[40]
		Resectable stage III-IVmelanoma	↑ better RFS	[42]
CD4^+^	Th1	melanoma	↑ Th1 genes expression	[46]
	Th17	MM was treated with Ipi ± HLA-A * 0201	↑ better RFS	[47]
	Treg	MM was treated with Ipi	↓ after treatment	[36]
	PD-1^+^ Treg	MM was treated with Pem	Significant ↓ after 1rst cycle was predictor of PFS and MSD	[68]
	Teff/Treg	MM was treated with Ipi	↑ was related to tumor necrosis	[59]
		MM was treated with Pem or Nivo	↑ Teff/Treg ratio after treatment	[66]
		MM was treated with combined anti-PD-1+anti-CTLA-4	↑ within the tumor	[54]
CD8^+^	Intratumoral CD8^+^	MM was treated with Pem	↑ density at the invasive margin in responders	[70]
	CD8^+^	MM was treated with Pem or Nivo	↑ PFS and ORR	[82]
	CD8^+^ Tc	MM was treated with anti-PD-1 alone or with anti-CTLA-4	↑ ORR	[71]
	CD8^+^ Teff	MM was treated with Ipi	↑ associated with DC	[28]
		stage IIIB-IV resectable melanoma treated with neoadjuvant anti-PD-1 plus anti-LAG-3	↑ intratumoral frequency in responders	[113]
DNTs	CD4^−^CD8^−^CD56^−^	MM was treated with anti-PD-1 alone or anti-CTLA-4	↓ in responders	[94]
Dysfunctional programs	TCF1^+^PD-1^+^CD8^+^	MM was treated with Nivo plus Ipi	↑ among responders and associated with longer response, OS, and PFS	[79]
	Ki67^+^PD-1^+^CTLA-4^+^CD8^+^	MM was treated with Pem	↑ after treatment	[80]
	CTLA-4^+^PD-1^+^CD8^+^	MM was treated with Pem or Nivo	↑ associated with better ORR and PFS	[83]
	Ki67^+^PD-1^+^CD8^+^	MM was treated with Pem	↑ Ki67^+^PD-1^+^CD8^+^ to tumor burden ratio associated with better OS, PFS, and ORR	[84]
	PD-1^hi^CTLA-4^hi^CD8^+^	MM was treated with anti-PD-1 plus anti-TIM-3	↑ better ORR, OS, and PFS	[85]
	Progenitor Tex/Terminally exhausted Tex	MM was treated with anti-PD-1	↑ ratio correlated to ORR	[79]

Abbreviations: ↓, decrease; ↑, increase; DC, disease control; Ipi, Ipilimumab; MSD, melanoma-specific death; Nivo, Nivolumab; MM, irresectable or metastatic; ORR, overall response rate; OS, overall survival; pts, patients; Pem, pembrolizumab;; PFS, progression-free survival; Ref, references; Tc, lymphocyte T cytotoxic; Tcm, lymphocyte T central memory; Teff, lymphocyte T effector; Tem, lymphocyte T effector memory; Tex, lymphocyte T exhausted; Th, lymphocyte T helper; Tn, lymphocyte T naïve; Treg, lymphocyte T regulatory.

## 4. Conclusions

From an oncologist’s perspective, while significant progress has been made in the systemic treatment of advanced melanoma, predictive biomarkers are urgently needed to better select responsive patients as early as possible in the course of treatment and improve their outcomes. The integration of lymphocyte T subset distribution determination, either PB or intratumoral, into clinical practice represents a promising avenue to personalize the ICI strategy and monitor its efficacy.

This is particularly relevant when it comes to determining their presence in PB peripheral blood, as it is accessible, allows longitudinal determinations, and may provide a more comprehensive understanding of the overall functioning of the immune system.

However, it should be borne in mind that differences in the impact on the T lymphocyte population may depend on the treatment applied in some situations. For example, we have mentioned differences between anti-PD-1 and anti-CTLA-4 monotherapy on Th1 cells and also compared them to anti-PD-1 and anti-CTLA-4 combinations.

Based on the existing research, Tcm and Tem within the memory compartment, especially within CD8^+^ T cells, would be crucial in inducing and maintaining a positive outcome of ICI, and there are data within all approved ICI strategies. Furthermore, within the CD8^+^ subset, Tc has been extensively associated with better ICI outcomes, particularly in anti-PD-1-based regimens, which cause its expansion and potentiate the cytotoxicity of these cells.

It is also important to highlight the important relationship between memory and exhaustion phenotypes and ICI outcome; Tex, particularly the progenitor subtype, are the ones more willing to reinvigorate on anti-PD-1-based treatment, and some transcription factors are crucial in promoting phenotype progression in both compartments, such as Tcf1 and eomes, and also the chemokine CXC3R1 that induces a T cell maduration within the memory phenotype. Higher levels of all of them have been correlated with better ICI outcomes, particularly anti-PD-1-based regimens.

Apart from the distribution of transcription factors and lymphocyte T subsets, we also found it important to mention some solid data on chemokines, in particular the IFN-γ-CXCL9/10-CXCR3 pathway, which, when induced by paracrine signals, promotes the migration of Th1, Tc and NK cells involved in the antitumor response to the TME and induces their activation. In contrast, IL-8 is a protumoral chemokine that promotes cancer cell survival and migration through multiple mechanisms.

However, the main obstacle to integrating these biomarkers into everyday clinical practice is the heterogeneity of the techniques and methodologies used in all the studies, flow cytometry, and other more sophisticated methods, such as scRNA-seq, most of which are now limited to research applications. However, as the number of clinical trials continues to grow, more and more parallel translation studies are being conducted to help homogenize these methodologies and advance them toward potential clinical implementation in the near future by providing evidence from prospective and randomized data.

Ultimately, the analysis of the distribution of T lymphocyte populations in PB by flow cytometry is a relatively simple, non-invasive method that can be performed at the bedside and provides results in a short time, making it a suitable biomarker to be incorporated into daily practice.

## Figures and Tables

**Figure 1 ijms-25-09506-f001:**
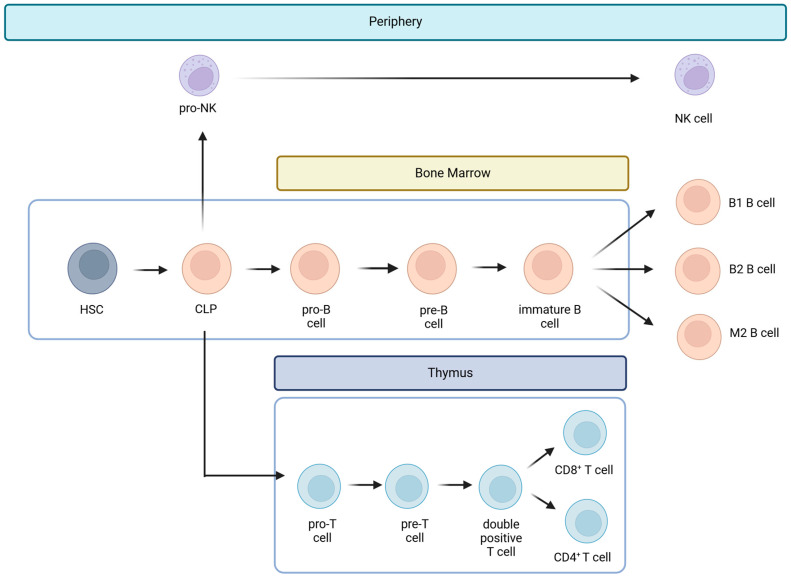
Lymphoid lineage cells. Hematopoietic stem cells (HSCs) give rise to different types of blood cells in lines called myeloid and lymphoid. Lymphoid cells include T cells, B cells, and natural killer cells (NK). The common lymphoid progenitor (CLP) is the precursor of the three lymphoid lineages and gives rise to pro-B cells, pro-T cells, and pro-NK cells. T cells mature in the thymus, and B cells in the bone marrow. On the other hand, pro-NK cells become NK cells on the periphery when a threat occurs.

**Figure 2 ijms-25-09506-f002:**
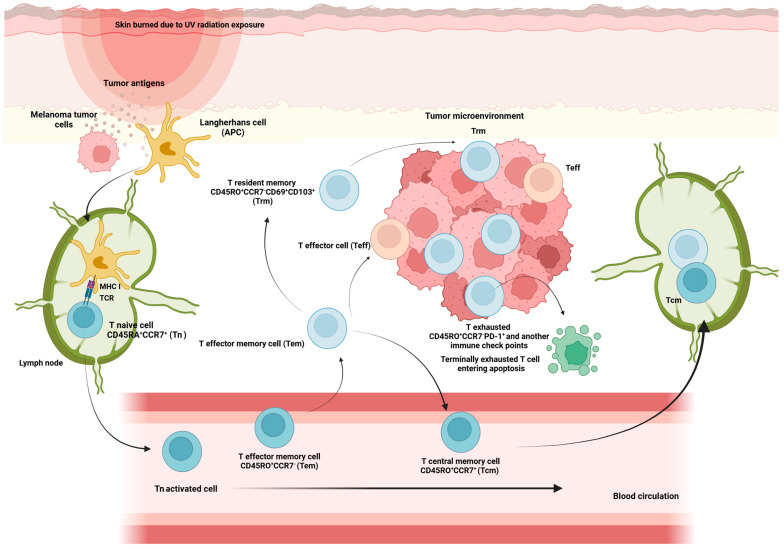
Memory compartment cells activation against melanoma tumor cells threat. Once antigen-presenting cells, such as Langerhans cells, detect a threat of melanoma tumor cells, they travel to the lymph node to find Tn cells. Tn cells bind to APC cells via the TCR-MHC I interaction, accompanied by a costimulatory signal. Through this process, they become activated and differentiate into effector cells (Teff) that migrate to inflamed sites and attack infected. After pathogen clearance, effector cells undergo apoptosis; the remaining T cells give rise to memory cells that have lost CD45RA expression and instead express CD45RO. Some Tn, when they become activated, differentiate into memory T cells with stem cells, resulting in an enhanced self-renewal capacity and multipotency to differentiate into all memory T cells. Also, a progressive maturation of T cells from Tn cells via Tcm and Tem to Teff cells is associated with chronic activation and shows features of dysfunction such as impaired cytokine secretion and the expression of markers of exhaustion, such as PD-1 (T exhausted CD45RO^+^CCR7^−^PD-1^+^). Some Tem differentiate into a population of memory CD8^+^ T cells, resident memory T cells (Trm), characterized for their persistence in tissue and non-recirculating in the blood.

**Figure 3 ijms-25-09506-f003:**
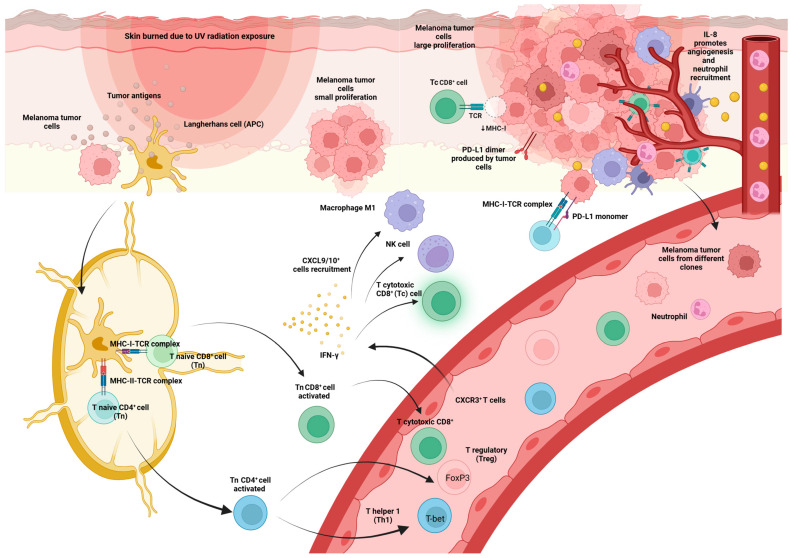
Distinct process of immune system activation against the threat of melanoma tumor cells. Once antigen-presenting cells (APC), such as Langerhans cells, detect a threat of melanoma tumor cells, they travel to lymph nodes to find Tn cells. Tn cells bind to APC cells via the TCR, accompanied by a costimulatory signal. Through this process, they become activated and differentiate into effector cells (Teff) that migrate to inflamed sites and attack infected. Tn CD4^+^ cells differentiate into Treg or Th1 depending on the transcription factor present in the original cell (FoxP3 or T-bet). Tn CD8^+^ cells differentiate into Tc CD8^+^ cells. CXCR3^+^ cells, which can be Th1 cells and effector CD8^+^ T cells, migrate into inflamed tissues and tumors and produce IFN-γ. IFN-γ stimulates CXCL9 and CXCL10^+^ cells, such as NK cells, Tc CD8^+^, and M1 macrophage recruitment. IL-8 secretion, which is increased in MM patients, causes neutrophil recruitment and stimulation of angiogenesis. At the tumor site, tumor cells produce negative immune checkpoints such as PD-L1, causing Tc CD8^+^ inactivation.

## Data Availability

Data is contained within the article.

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
