# Peer review of "Lymphocyte T Subsets and Outcome of Immune Checkpoint Inhibitors in Melanoma Patients: An Oncologist’s Perspective on Current Knowledge"

_ijms, 2024, doi:10.3390/ijms25179506_

Round 1
Reviewer 1 Report
Comments and Suggestions for Authors
Although the work is interesting and provides extensive information on melanoma, it is presented in a rather confusing manner. For instance, there is no proper introduction to melanoma; it is only briefly mentioned in the abstract. The review is lengthy and lacks a substantial conclusion. The conclusion is sparse in content compared to the wealth of information provided in the main text.
Reviewer 2 Report
Comments and Suggestions for Authors
Your paper entitled -Lymphocyte subsets and immunotherapy responsiveness in melanoma patients: the oncologist’s perspective- presents an important scientifically subject about known biomarkers to show their implications into prognostic or diagnostic, but remain to improve the quality of your manuscript by following next issues:
-1. About your title, please change it to highlights the correlation between objective, results, discussion with more impact about the importance of known lymphocyte biomarkers in your study.
-2. Please revise entire abstract to highlights your aim that propose you in the ending of the abstract –lines 26 to 28. Please make difference between “lymphoid subsets” than “Lymphocyte subsets” from title.
3-In keywords section, please adapt this section with entire manuscript, because “immune checkpoint inhibitors; lymphoid subsets” are many but in your abstract presented only genetically biomarkers (BRAFV600) for advanced melanoma cases. Please revise to establish the importance of your study in correlation with international scientifically references about this subject.
4- About your graphically abstract, I suggest moving in Introduction section and explain better about your biomarkers, about methodology because you have TILs or PDL-1 from tissue as histopathological analyses and PBMCs biomarkers by flow cytometry to lymphocytes and cytokines patterns, and responsiveness to immunotherapy. I suggest diving the biomarkers involved in diagnostic establishment and biomarkers involved in responsiveness to immunotherapy by tissue or blood samples and specifically methodology to have a general figure. Please, establish clearly the biomarkers that are characteristically for responsiveness to immunotherapy and others to inflammation or cancer. Also, please specify Tregs, double negative T, IL-8 cytokines cell population.
5- Please, revise the scope of your manuscript (lines 86-92) to be in correlation with the entire manuscript. Also, specify if used only advanced melanoma cases by BRAFV600 diagnostic in this review to see the Lymphocytes pattern involved in immunotherapy responsiveness (ICIs) or the manuscript treats the subject in generally way?
6- Please, add, the Results section, before beginning of 2. Lymphoid populations and outcomes of melanoma patients treated with ICI.
7- Please, in entire manuscript add the punctuation marks or the free spaces in paragraph to improve the quality of the manuscript. All terms “in vivo”, please add in italic mode. Please add the scientifically explanation for all terms. Please accords more attention to writing this review.
Comments on the Quality of English LanguageMinor editing of English language required.
